# Pushed to extremes: distinct effects of high temperature versus pressure on the structure of STEP

Liliana Guerrero[1,2,7], Ali Ebrahim[1,7], Blake T. Riley [1], Minyoung Kim [1,3], Qingqiu Huang [4], Aaron D. Finke[4] & Daniel A. Keedy [1,5,6✉]

Protein function hinges on small shifts of three-dimensional structure. Elevating temperature or pressure may provide experimentally accessible insights into such shifts, but the effects of these distinct perturbations on protein structures have not been compared in atomic detail. To quantitatively explore these two axes, we report the first pair of structures at physiological temperature versus. high pressure for the same protein, STEP (PTPN5). We show that these perturbations have distinct and surprising effects on protein volume, patterns of ordered solvent, and local backbone and side-chain conformations. This includes interactions between key catalytic loops only at physiological temperature, and a distinct conformational ensemble for another active-site loop only at high pressure. Strikingly, in torsional space, physiological temperature shifts STEP toward previously reported active-like states, while high pressure shifts it toward a previously uncharted region. Altogether, our work indicates that temperature and pressure are complementary, powerful, fundamental macromolecular perturbations.

[1] Structural Biology Initiative, CUNY Advanced Science Research Center, New York, NY 10031, USA. [2] PhD Program in Biochemistry, CUNY Graduate Center, New York, NY 10016, USA. [3] Department of Molecular Biology, Princeton University, Princeton, NJ 08544, USA. [4] Cornell High Energy Synchrotron Source (CHESS), Cornell University, Ithaca, NY 14853, USA. [5] Department of Chemistry and Biochemistry, City College of New York, New York, NY 10031, USA. [6] PhD Programs in Biochemistry, Biology, & Chemistry, CUNY Graduate Center, New York, NY 10016, USA. [7]These authors contributed equally: Liliana Guerrero, Ali Ebrahim. ✉email: dkeedy@gc.cuny.edu

The biological functions of many proteins require transitions between conformational substates[1–3]. Despite their functional importance, protein conformational substates are often difficult to characterize. X-ray crystallography can prove useful in this regard by revealing alternate conformations that coexist in crystals at partial occupancy, as shown by electron density maps[4]. Such conformational ensembles can be shifted by discrete, localized, targeted perturbations like ligands[5] or mutations[6], revealing insights into allostery and enzyme catalysis. However, known ligands are unavailable for most sites in most proteins, and predicting the effects of mutations is difficult. By contrast, continuous, global, generic biophysical perturbations offer advantages: they can be applied to any protein, affect the entire structure simultaneously, and can be titrated to shift conformational distributions and map correlated conformational changes relevant to function[7].

One such biophysical perturbation, which has gained traction as a valuable experimental variable in structural biology and biophysics, is temperature (T). Room-temperature (RT) X-ray crystallography[8] avoids structural biases of cryogenic-temperature crystallography, revealing differences in protein conformation[9–12], ligand binding[13,14], and solvation layers[10,14]. Multitemperature crystallography provides additional insights into conformational coupling[11,12,15]. Notably, crystal structures at physiological temperature (37 °C, 310 K) can reveal unique protein conformations[12,16]. RT crystallography methods are rapidly improving[8], including serial crystallography[17]. RT crystal structures are also increasingly used in computational simulations[18–20].

Complementary to temperature, but relatively underexplored, is pressure (P). Whereas high temperature stabilizes states with high entropy, high pressure stabilizes states with low volume, isolating distinct excited states that may have unique links to biological function[21–25]. Importantly, pressure-induced structural changes on the sub-angstrom level observed by high-pressure X-ray crystallography have been shown to be directly related to protein function[26]. Other past high-pressure protein crystallography studies showed non-uniform responses of coordinates and B-factors[27], non-compressive conformational shifts mirroring those induced by pH change[28], water infiltration into engineered[29,30] and natural cavities[31], crystal phase transitions[31,32], conformational shifts of functional residues in an allosteric network[32], and changes in ligand affinity[33].

Despite this foundation, relatively few studies have explored the detailed effects of pressure on protein conformational ensembles using crystallography. While a few studies have highlighted protein alternate conformations for isolated residues[31,32], to our knowledge no study has comprehensively explored the effects of pressure on detailed conformational ensembles of all residues throughout a protein structure. Moreover, very few studies[28] have compared the atomic-level effects of elevated temperature vs. pressure on protein crystal structures. It thus remains unclear whether, and how, these two fundamental thermodynamic perturbations differentially affect protein conformational ensembles, which limits our toolkit for probing fundamental connections between conformational heterogeneity and biological function.

An attractive system to investigate the differential effects of temperature vs. pressure on protein conformational ensembles is the protein tyrosine phosphatase (PTP) enzyme STEP (PTPN5). STEP is a brain-specific PTP, and a validated therapeutic target for Alzheimer's disease[34], Fragile X syndrome[35], and Parkinson's disease[36]. The public Protein Data Bank[37] includes 8 high-resolution (1.66–2.15 Å) crystal structures of STEP (7 human, 1 mouse) with different ligands, demonstrating its tractability with crystallography. Of these structures, 3 are in an inactive-like state,

either bound to a competitive inhibitor[38] or inactivated through the acetylation of the catalytic cysteine[39], while another 3 are in an active-like state, either bound to an allosteric small-molecule activator[40] or in a Michaelis-like complex with a pTyr substrate bound to a catalytic C472S mutant[41]. As revealed in these structures, STEP has several unusual features among PTPs, including an atypically open active-site WPD loop conformation[41] and an allosteric site with a small-molecule activator (not inhibitor)[40]. However, all existing STEP structures are at cryogenic temperature and ambient pressure.

Here we report high-resolution (<2 Å) crystal structures of unliganded STEP at high temperature (HiT) and at high pressure (HiP), along with a reference structure at low temperature and low pressure (LoTP). To our knowledge, these new structures of STEP represent several firsts. Our high-temperature structure is only the eleventh crystal structure of any protein, and the first of any phosphatase, at physiological temperature or above (≥310 K). Our high-pressure structure of STEP is also the first of any phosphatase at high pressure. Together, our new structures make STEP the first protein with crystal structures at both physiological temperature and high pressure, presenting a unique opportunity to compare the effects of these two distinct perturbations on protein conformational ensembles.

By quantitatively interrogating these data, we reveal that temperature and pressure have complementary effects on the conformational landscape of STEP. These two perturbations have opposite effects on the crystal lattice but surprisingly similar effects on the protein molecular volume, stabilize distinct ordered water molecules throughout the protein, induce backbone shifts in non-overlapping regions of the structure, and rearrange different sets of side chains. We observe a previously unseen arrangement of product-like anions in the active-site pocket, new conformations of conserved catalytic residues only at high temperature, and an active-like conformation of an active-site loop only at high pressure. Surprisingly, using a new computational method for analyzing distributions of protein structures[42], we find that high temperature in the unliganded state induces a coordinated global shift toward previous ligand-bound active-like structures, whereas high pressure shifts the protein toward a previously unseen region of conformational space. Overall, our results illustrate the potential of manipulating protein structures with a broad spectrum of physical perturbations to gain unique insights into their mechanical coupling and biological function.

## Results

**X-ray datasets at high temperature vs. pressure.** To compare the effects of temperature vs. pressure on the STEP catalytic domain, we used similarly prepared crystals to obtain three complementary crystal structures: one at cryogenic temperature (100 K) and ambient pressure (0.1 MPa), one at physiological temperature (310 K) but ambient pressure, and one at high pressure (205 MPa) but cryogenic temperature via high-pressure cryocooling[43]. For the remainder of this paper, we refer to the structure at low temperature and low pressure as LoTP, the structure at high temperature as HiT, and the structure at high pressure as HiP. The diffraction datasets and resulting refined structures were of high quality, including acceptably similar resolutions (Table 1). Therefore, these datasets can be directly compared to gain insights into the differential effects of temperature vs. pressure on the conformational ensemble of STEP.

All three structures have the expected PTP catalytic domain architecture (Fig. 1a, b), including several key loops surrounding the active site region (Fig. 1c). Broadly speaking, they are similar to the 7 previously published structures of human STEP (8 including 1 structure of mouse STEP; Supplementary Fig. 1).

## Table 1 Crystallographic statistics.

| PDB ID | 8SLS | 8SLT | 8SLU |
|---|---|---|---|
| Dataset name | LoTP | HiT | HiP |
| Data collection[a] | | | |
| Temperature (K) | 100 | 310 | 100 |
| Pressure (MPa) | 0.1 (ambient) | 0.1 (ambient) | 205 |
| Space group | P2₁2₁2₁ | P2₁2₁2₁ | P2₁2₁2₁ |
| Cell dimensions | | | |
| $a, b, c$ (Å) | 39.67, 63.51, 135.16 | 39.98, 64.49, 137.21 | 39.14, 63.47, 134.20 |
| $a, \beta, \gamma$ (°) | 90, 90, 90 | 90, 90, 90 | 90, 90, 90 |
| Resolution (Å) | 1.71–67.58 | 1.96–68.61 | 1.84–57.37 |
| $R_{merge}$ | 0.108 (2.37)[b] | 0.170 (2.398) | 0.114 (2.127) |
| $I / \sigma I$ | 10.23 (0.59) | 7.79 (0.63) | 10.71 (0.64) |
| Completeness (%) | 98.64 (93.42) | 99.80 (99.03) | 99.45 (99.12) |
| Redundancy | 4.7 (4.8) | 6.7 (6.9) | 6.6 (6.6) |
| Refinement | | | |
| Resolution (Å) | 1.71–67.58 | 1.96–68.61 | 1.84–57.37 |
| No. reflections | 37,552 (3655) | 26,353 (2573) | 29,880 (2949) |
| $R_{work}/R_{free}$ | 19.15/22.44 | 17.44/20.57 | 19.48/23.66 |
| No. atoms | 4871 | 4721 | 4920 |
| Protein | 4700 | 4643 | 4799 |
| Ligand/ion | 23 | 10 | 23 |
| Water | 149 | 69 | 98 |
| B-factors | 41.46 | 51.61 | 45.55 |
| Protein | 41.45 | 51.71 | 45.63 |
| Ligand/ion | 34.43 | 43.79 | 41.57 |
| Water | 42.65 | 46.43 | 42.34 |
| R.M.S. deviations | | | |
| Bond lengths (Å) | 0.01 | 0.02 | 0.01 |
| Bond angles (°) | 1.20 | 1.40 | 1.24 |

[a]One crystal was used for each dataset.
[b]Values in parentheses are for the highest-resolution shell.

However, the unit cell dimensions differed from the LoTP reference dataset in opposing ways: at HiT the unit cell volume expanded by 3.9%, whereas at HiP the unit cell was instead compressed by 2.1% (Table 2). Comparatively, the protein molecule itself was more robust, but was still affected by temperature and pressure: at HiT the protein volume expanded by 1.1%, whereas at HiP it still expanded, but by only 0.4%. Thus elevated temperature expands the unit cell and, to a lesser extent, the protein itself; by contrast, elevated pressure compresses the unit cell, but still allows the protein itself to slightly expand. These observations suggest that temperature vs. pressure has more complex effects on STEP than might be naïvely expected from the unit cell changes alone. Indeed, these distinct perturbations induce a variety of conformational changes distributed throughout the structure of STEP, including some with potential biological relevance, as shown below.

**Unique arrangement in STEP active site**. One notable feature of our new structures differs from the previous STEP structures: the active site binds two sulfate molecules (Fig. 1c, d and Supplementary Fig. 2a). The top sulfate sits just beneath the catalytic WPD loop, where a lone sulfate has been observed previously in PDB ID 2bv5, 2bij[39], and 6h8r[40], and inhibitors with negatively charged moieties have been observed in PDB ID 5ovr, 5ovx, and 5ow1[38] (Supplementary Fig. 2b).

The bottom sulfate is well-coordinated by the catalytic P loop, analogous to the phosphate group in the phosphotyrosine (pTyr) substrate in PDB ID 2cjz[41] (Supplementary Fig. 2c). A phosphocysteine reaction intermediate with a covalent bond to Cys472 is an unlikely explanation of our data, as refining such a putative model resulted in strong negative difference density peaks (Supplementary Fig. 3), our crystals contained high concentrations of lithium sulfate but not any phosphate-containing compounds, and previous intermediate-bound PTP structures used inactivating mutations to capture such

intermediates[41,44] whereas our structure is wildtype. In our structures, as supported by strong electron density, Cys472 predominantly adopts a side-chain rotamer that points away from the sulfate, thus avoiding a steric clash. This rotamer in our new structures is rare for STEP: it was previously only seen as a partial-occupancy alternate conformation in an allosterically activated structure (PDB ID 6h8r) (Supplementary Fig. 2b). Further supporting this primary rotamer, Ringer curves[4] for Cys472 for all three datasets have a dominant peak for the χ1 side-chain dihedral angle near 180°. In addition, Ringer curves from different model preparations for all three datasets also have a secondary χ1 peak near +60°, suggesting sensitivity of Ringer to precise input coordinates and/or subtle sensitivity to our global perturbations (Supplementary Fig. 4). This secondary Ringer peak is consistent with an alternate rotamer conformation for Cys472, which is further supported by unbiased Polder[45] (Supplementary Fig. 5) and omit (Supplementary Fig. 6) electron density maps. Moreover, the backbone density for Cys472 is consistent with multiple positions, suggesting Cα displacements that are perpendicular to the chain direction (Supplementary Figs. 5 and 6) and similar in magnitude (0.44–0.93 Å) to those seen between alternate conformations in a previous structure of STEP (0.73 Å for PDB ID 2bv5, although Cys472 is acetylated in that structure and does not change rotamer). Taken together, these observations support the interpretation that Cys472 samples both a rare primary rotamer and a low-occupancy alternate rotamer that is sterically mutually exclusive with a fortuitously observed sulfate bound at high but non-unity occupancy.

Thus, although previous structures of STEP have sulfates or phosphate-like chemical groups independently in each of these sites, no previous structure has them in both sites simultaneously. The closest comparison is PDB ID 2bv5, in which the catalytic Cys472 is modeled as acetylated in the bottom site and a sulfate is in the top site (Supplementary Fig. 2d), but this arrangement differs in chemical character from what we observe.

**Alterations to ordered solvent**. In addition to global changes to the crystal lattice, high temperature, and pressure have striking effects on the solvation layer surrounding the protein. In our manually modeled, deposited structures, LoTP has by far the most waters, HiT has by far the fewest, and HiP has an intermediate number (Supplementary Table 1 and Fig. 2). To confirm that this result is not due to the small differences in resolution between datasets (Table 1), we truncated the LoTP and HiP diffraction data to match the resolution of the HiT dataset (1.96 Å), then performed fully automated, unbiased water placement for all three structures (see "Methods" section). The resulting water counts are similar to the counts of manually placed waters in our deposited structures (Supplementary Table 1), thus validating the conclusions derived from the latter.

Turning to specific water positions in our deposited structures, 17 (24.6%) of the HiT waters and 23 (23.5%) of the HiP waters were distinct from any LoTP water (> 2 Å, accounting for crystal symmetry) (Fig. 2). Of these 40 new positions, only 1 (2.5%) is common to both HiT and HiP. This suggests that high temperature and high pressure do not merely retain a subset of ordered waters, but rather stabilize new water positions, resulting in a distinct pattern of solvation. As shown below, some of these unique waters are located at functional sites in STEP (Fig. 3). In total, we reveal 67 (LoTP) + 16 (HiT) + 22 (HiP) = 105 waters that are unique to one structure (Fig. 2), further underscoring the value of crystallography with different axes of perturbations for mapping accessible patterns of protein solvation.

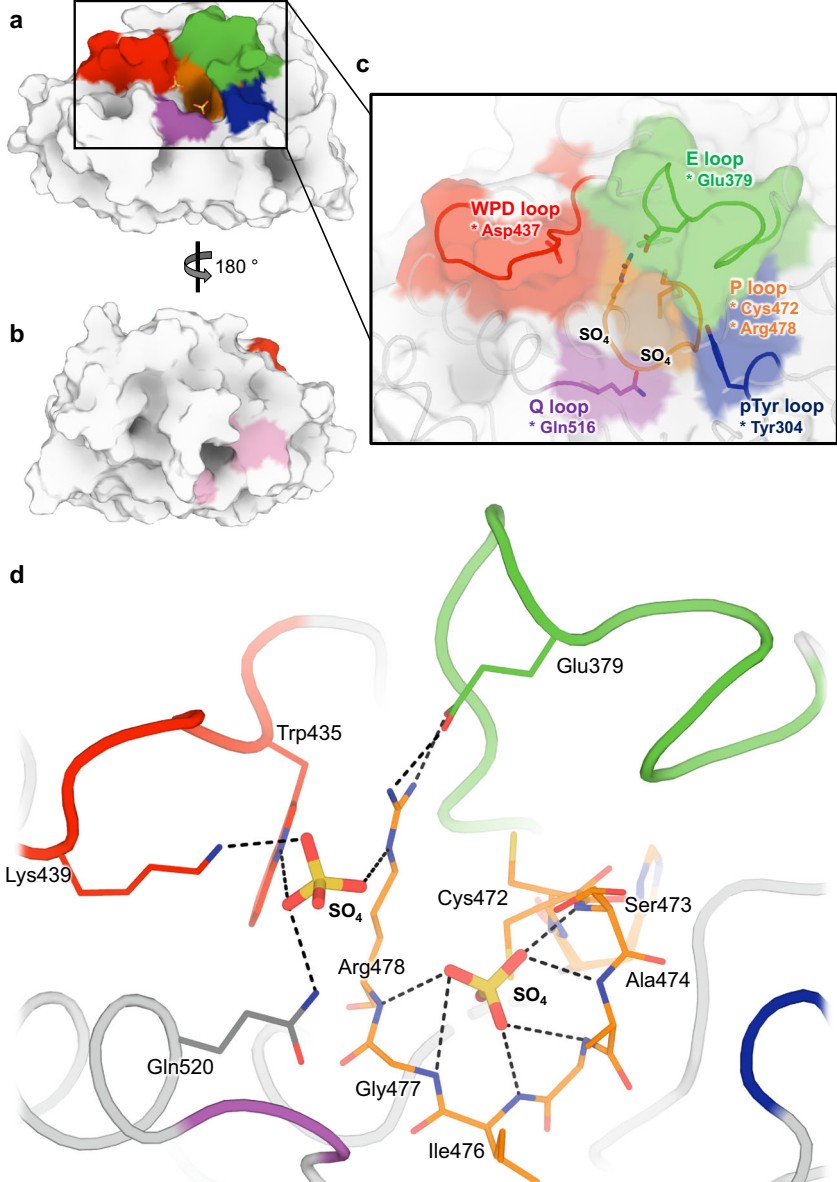

**Fig. 1 Structural overview of STEP, including two sulfates in active site. a** Overview of STEP catalytic domain, centered on active site. **b** 180° rotation of (**a**) to show allosteric activator binding site[40], with key residues highlighted in pink. **c** Zoom-in of (**a**) showing several key active-site loops and two sulfates bound in the active site cleft. Key catalytic residues are denoted with an asterisk. **d** Interactions between two sulfates and nearby residues in the active site of our LoTP structure.

**Table 2 Change in unit cell and protein volume at high temperature vs. high pressure.**

|  |  | LoTP | HiT | HiP |
|---|---|---|---|---|
| Unit Cell | a (Å) | 39.67 | 39.98 (+0.8%) | 39.15 (−1.3%) |
|  | b (Å) | 63.51 | 64.49 (+1.5%) | 63.45 (−0.1%) |
|  | c (Å) | 135.16 | 137.21 (+1.5%) | 134.22 (−0.7%) |
|  | Cell Volume (Å$^3$) | 340527.7 | 353769.9 (+3.9%) | 333411.5 (−2.1%) |
| Protein | Protein Volume (A$^3$) | 37632.6 | 38039.1 (+1.1%) | 37783.3 (+0.4%) |

Absolute number given first (% change relative to LoTP given in parentheses. Protein total volume calculated by the ProteinVolume software[72].

**Global effects on protein conformation**. To explore differential effects of high temperature vs. pressure on the protein molecule itself, we examined Cα displacements in the HiT and HiP structures relative to the reference LoTP structure. A global plot of this Cα distance vs. amino acid sequence (Fig. 3a) reveals that most regions are similar in the three structures, with Cα distances <0.3 Å, but several local regions shift relative to the reference structure. These shifts tend to occur either only at high temperature or only at high pressure, suggesting that the protein responds to these different perturbations in distinct ways.

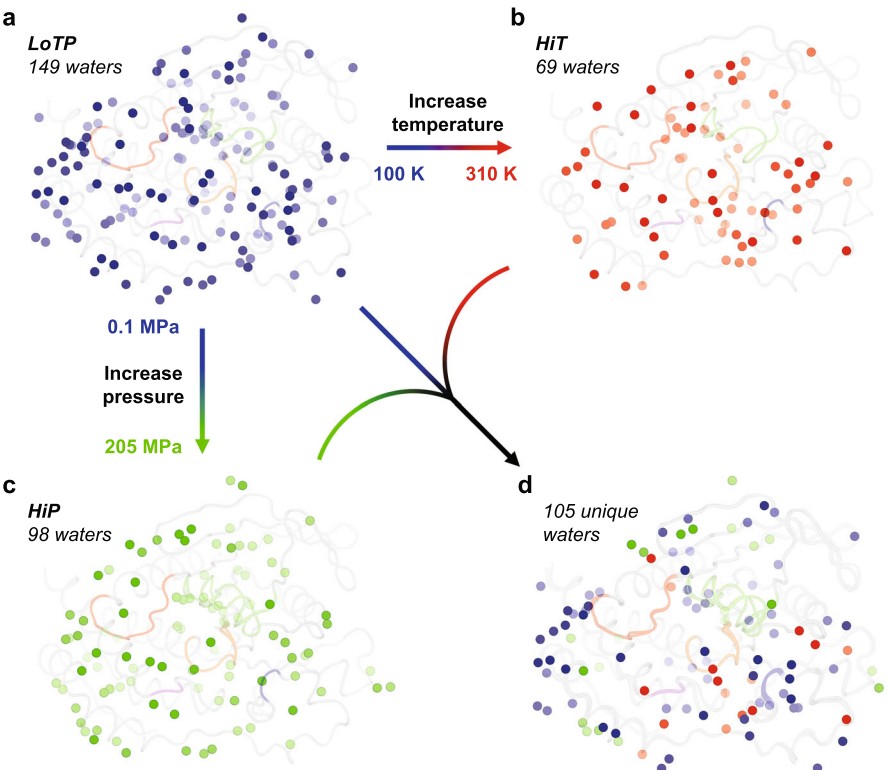

**Fig. 2 Ordered water molecules are sensitive to temperature and pressure. a**–**c** All ordered water molecules at (**a**) LoTP, (**b**) HiT, and (**c**) HiP are shown. **d** Only the waters unique to each structure, i.e. >2 Å from any water in the other two structures. Coloring for active-site loops as in Fig. 1.

Our structures were obtained in the same crystal form as PDB ID 2bv5, which, like our LoTP structure, is a cryogenic-temperature, ambient-pressure dataset. Cα distance analysis shows that for many key regions, 2bv5 is similar to our LoTP structure, whereas our HiT and HiP structures are more different (Supplementary Fig. 7). Thus the effects of temperature and pressure are generally greater than the variability inherent to determining structures of the same protein in similar conditions by different scientists at different times.

Beyond 2bv5, all other previous human STEP structures were in a different crystal form (same space group but longer a and shorter c axes). These exhibit similar or greater Cα distances than do our HiT and HiP structures at several sites in STEP (Supplementary Fig. 7b). All previous STEP structures were determined at cryogenic temperature and ambient pressure. This indicates that, at least at a gross level, differences in crystal contacts may elicit protein structural variability[46] that encompass much of the variability elicited by experimental perturbations such as temperature and pressure. Nonetheless, as shown below, temperature and pressure each induce unique conformational states of STEP.

**Local effects on key structural regions**. To explore the basis of these global structural differences, we examined several local areas with distinct conformations in the HiT vs. HiP structures. One local region that responds strongly to pressure — but not to temperature — is the E loop (Fig. 3a region (iii)). The E loop of PTPs, containing several Glu (E) residues, is located adjacent to the catalytic WPD loop and P loop (Fig. 1). Among previous structures of STEP, the E loop exhibited substantial variability (Supplementary Figs. 1 and 7). The two main states previously modeled for this loop were the inactive-like state in 2bv5 (with an acetylated catalytic Cys472), and the active-like state in 6h8r (bound to a distal allosteric small-molecule activator). To validate

these previous models, we inspected the electron density maps for all previous STEP crystal structures (7 human, 1 mouse). We determined that all of these structures besides 6h8r were either already modeled with a 2bv5-like conformation, or were unmodeled but could be better explained by a 2bv5-like conformation than by a 6h8r-like conformation. Thus, the active-like state of the E loop was only legitimately observed in the allosterically activated structure 6h8r, even though the density was somewhat noisy (Supplementary Fig. 8).

In contrast to previous STEP structures, our HiP electron density for the E loop, albeit also noisy, is consistent with the presence of both an inactive-like state as in 2bv5 and an active-like state as in 6h8r. We therefore modeled both states as alternate conformations (Fig. 4a, b). Deletion of either of these conformations and calculation of omit maps results in positive Fo-Fc difference density peaks for the omitted model (Fig. 4c, d), suggesting both are present. The 6h8r-like conformation exists in our HiP structure despite having a different crystal form than 6h8r. By contrast to HiP, our crystallographically isomorphous LoTP and HiT structures are essentially identical to 2bv5 for the E loop. Therefore, high pressure appears to uniquely stabilize a conformation of a key active-site loop that is correlated with an allosterically activated state of human STEP.

Another region that responds to pressure is residues 287-306, encompassing the pTyr loop, also known as the substrate-binding loop (SBL) (Fig. 3a, b region (ii)). Backbone shifts in this region play a crucial role in defining the depth of the catalytic pocket[47]. In our models, the backbone of this region, particularly the N-terminal portions, shifts >1 Å from LoTP to HiP (Fig. 3c). In addition, the backbone of the C-terminal portions of this region, corresponding to the pTyr loop itself, shifts by up to ~0.7 Å from LoTP to HiT (Fig. 3c). Notably, the backbone for the pTyr loop residue Tyr304, whose side chain directly interacts with and helps position the pTyr substrate during catalysis, shifts at both HiP

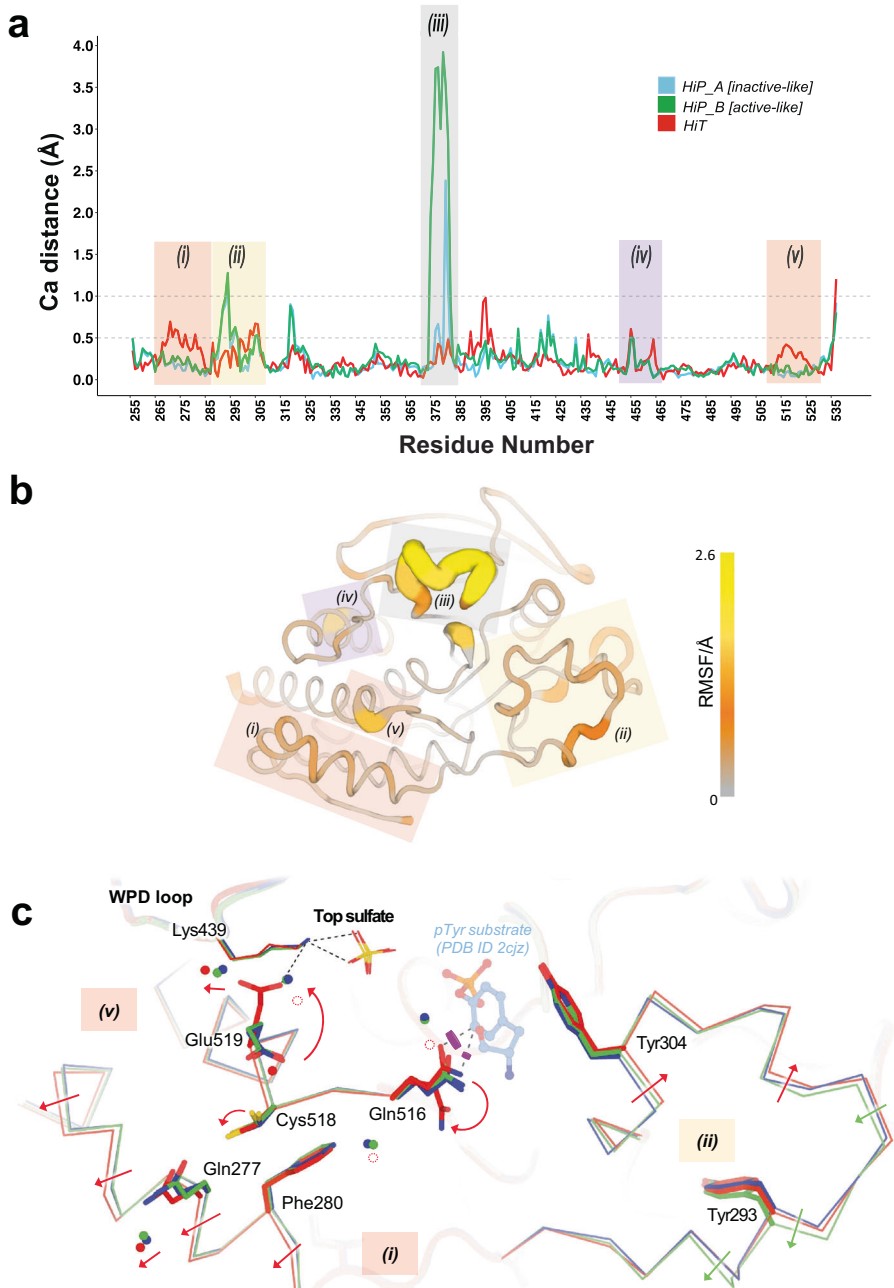

**Fig. 3 Global backbone displacements due to high temperature vs. pressure. a** Cα distances for the HiT and HiP structures relative to the reference LoTP structure are plotted vs. amino acid sequence. The two alternate conformations for the E loop in the HiP structure are separated, although both have high Cα distance to LoTP. See also Supplementary Fig. 3. Regions with interesting backbone differences are highlighted; those highlighted with the same color are adjacent in the tertiary structure. The data for generating this graph is available in Supplementary Data 1. **b** Structure of STEP with color and cartoon width corresponding to Cα root-mean-square fluctuation (RMSF) between our HiT, HiP, and LoTP structures. Same highlighted regions as in **a**. **c** Zoom-in of active-site area including region (i) (residues 267–282, α1′-α2′ helices), region (v) (residues 515–531, Q loop), and region (ii) (residues 287–306, pTyr loop). Catalytic WPD loop and top sulfate are shown nearby. Magenta disks show putative steric clashes between Gln516 and an aligned pTyr substrate from PDB ID 2cjz (not in our structures) which is included for context. See Fig. 4 for zoom-in of region (iii), and Supplementary Fig. 9 for zoom-in of region (iv).

and HiT, yet its side chain remains in place. Overall, these observations suggest a degree of plasticity in the substrate-binding region, which we speculate may help accommodate different pTyr-containing substrates.

In contrast to these regions that respond to pressure, other regions of STEP respond only to temperature. The backbone of the α1′-α2′ helical region near the N-terminus (residues 266–284)

shifts by up to ~0.7 Å at HiT but not HiP (Fig. 3a, b region (i)). In addition, the junction between the active-site Q loop and the α6 helix (residues 514–524) shifts by up to ~0.4 Å, also at HiT but not HiP (Fig. 3a, b region (v)). These new HiT conformations differ not only from our HiP and LoTP structures, but also from the only previous STEP structure with the same crystal form, 2bv5, which was at cryogenic temperature (Supplementary Fig. 7a).

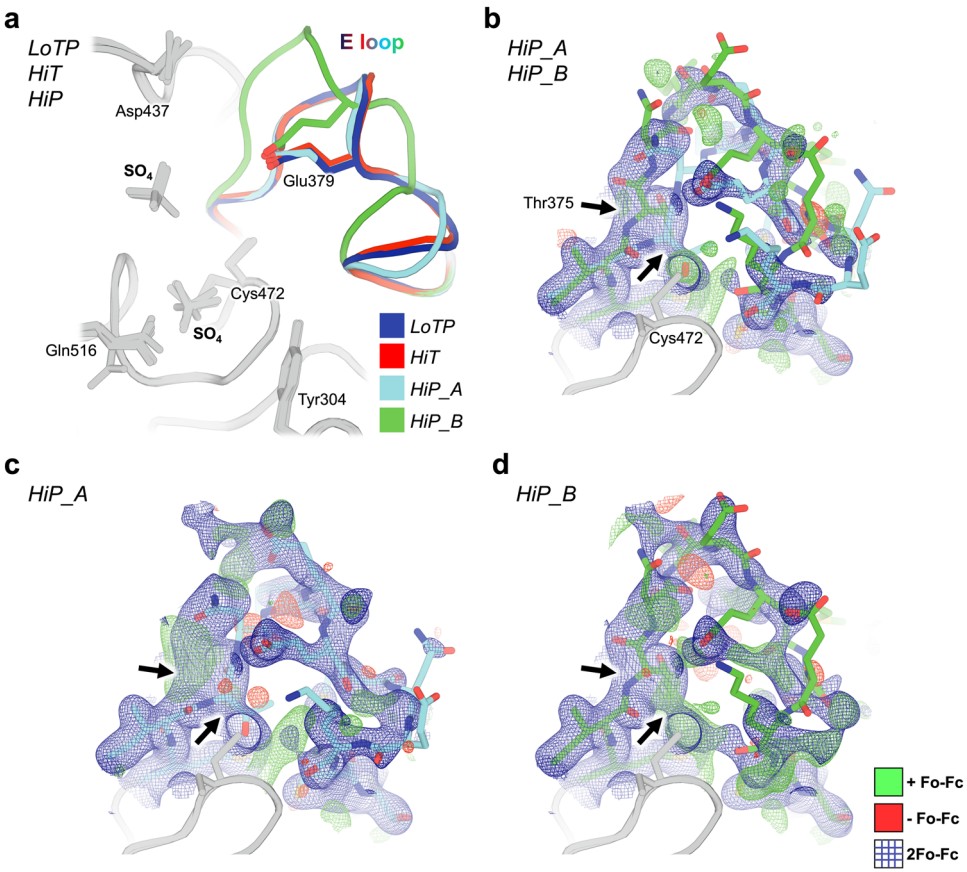

**Fig. 4 The E loop is reorganized uniquely at high pressure. a** Overlay of all three new STEP structures. The conformation of the E loop is nearly identical at LoTP (blue) and HiT (red), but deviates into two distinct conformations at HiP (cyan, green). Glu379 remains within the same conformational space regardless of E-loop conformation. **b** HiP dual E loop (cyan, green sticks) with 2Fo-Fc (blue mesh, 1 σ) and Fo-Fc difference (green mesh, +3 σ; red mesh, -3 σ) electron density maps. Thr375 is the first full amino acid where the E loop completely separates into two distinct conformations (black arrows). **c** HiP_A conformation of E loop with 2Fo-Fc and Fo-Fc difference electron density maps, omitting the HiP_B state. Arrows indicate Thr375 deviation. **d** HiP_B conformation of E loop with 2Fo-Fc and Fo-Fc difference electron density maps, omitting the HiP_A state. Arrows indicate Thr375 deviation.

These backbone shifts are coupled to other notable changes to side-chain conformational ensembles (Fig. 3c). In concert with the Q loop backbone shift in this interface, the side chain of Cys518 (from the Q loop) switches from two rotamers to one. The disappearance of the alternate rotamer for Cys518 eliminates a hydrogen bond to the adjacent Glu519, causing the latter to switch to a new rotamer (see also Fig. 5b). The new Glu519 rotamer engages in a previously unseen interaction with Lys439 from the catalytic WPD loop, which coordinates the top sulfate (Fig. 3c). This conformation of Glu519 is not present in any previous STEP structures: it is unique to our HiT structure. Several of these changes are also correlated with shifts or disordering of nearby water molecules (Fig. 3c), illustrating an interplay between protein and solvent structure.

The Q loop backbone shift is also correlated with an alternate side-chain rotamer for Gln516 (Fig. 3c) that has only been seen in two previous structures: bound to a pTyr substrate (2cjz; Supplementary Fig. 2c), and bound to a distal allosteric activator (6h8r; Supplementary Fig. 9). In particular, this new rotamer avoids what would otherwise be a steric clash with the pTyr substrate, which binds immediately adjacent to Gln516 (Fig. 3c). These observations suggest that HiT may capture an active-like conformation of STEP, even in the unliganded form, that is more compatible with formation of the Michaelis complex. Interestingly, Gln516 is immediately adjacent to Ile515, which is the only residue in STEP to have an unavoidable but real Ramachandran outlier — consistent with previous observations that validated,

geometrically strained residues, while rare, occur preferentially at active sites[48].

In the context of the crystal lattice, α1′-α2′ also abuts the distal S loop (residues 462-465), parts of which shift by >0.5 Å at HiT but not HiP (Supplementary Fig. 9). Interestingly, the S loop forms part of the binding site for a class of allosteric small-molecule activators that are unique to STEP[40] (Supplementary Fig. 9). This coincidence of temperature-sensitive regions in 3D space suggests that subtle lattice expansion at elevated temperature can allow a protein "breathing room" to adopt subtly different conformations, including at functionally important regions.

**Effects on pockets and cavities**. To assess how temperature vs. pressure affect various packing defects in the structure of STEP, we used the program CASTp[49] to measure the volumes of all pockets and/or cavities in each structure (Supplementary Fig. 10). Interestingly, the largest pocket in each of the three structures was the allosteric activator site. Relative to the LoTP structure (128.7 Å³), the volume of this pocket increased at HiT (140.0 Å³) but decreased at HiP (74.8 Å³), consistent with general expectations of expansion with increasing temperature and compression with increasing pressure.

Considering all pockets/cavities in each structure, the mean volume relative to LoTP (7.8 Å³) increases for HiT (11.3 Å³) and slightly decreases for HiP (7.5 Å³). However, relative to LoTP, we observe no statistically significant difference in the distribution of

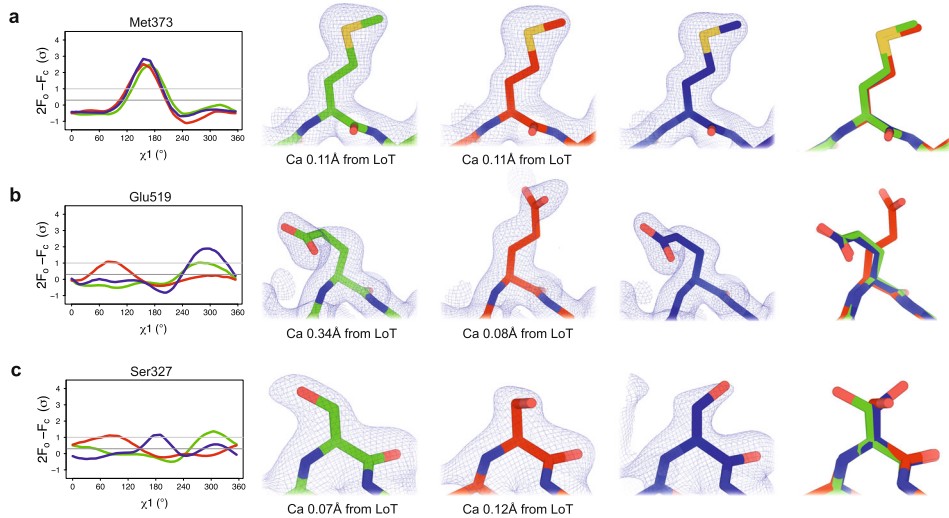

**Fig. 5 Examples of different side-chain conformations at high temperature and/or pressure.** For each example residue, the following panels are shown: (*Left*) Overlaid Ringer curves for our three datasets[4]. (*Middle*) Our three structures with 2Fo-Fc (contoured at 1 σ) and Fo-Fc (contoured at ±3 σ) density maps. (*Right*) Our three structures overlaid. LoTP in blue, HiT in red, HiP in green. Examples: (**a**) Met373 has the same χ1 peak (*t*, near 180°) for LoTP, HiT, and HiP. **b** Glu519 has similar χ1 peaks for LoTP and HiP (*m*, near -60°), but a different peak for HiT (*p*, near +60°). **c** Ser327 has different χ1 peaks for LoTP (*t*), HiT (*p*), and HiP (*m*). χ1 rotamer nomenclature from[50]. The data for generating the Ringer curves in **a–c** is available in Supplementary Data 2.

these volumes for either HiT or HiP (Welch's two-sample *t*-test, $p = 0.598$ and $p = 0.951$, respectively). Thus, although some individual pockets react differently to different perturbations, at least by some measures the overall packing in the protein is not dramatically different.

**Widespread changes to torsion angles**. We next examined in detail how temperature vs. pressure affected conformations throughout the entirety of the STEP catalytic domain, using torsion angles in several ways. First, we performed Ringer analysis for each side chain in each structure by rotating around the Cα-Cβ vector (χ1 torsion angle) and measuring the 2Fo-Fc electron density value at each possible γ heavy atom position[4]. For each residue, we then calculated a correlation coefficient (CC) between Ringer curves for each pair of datasets[15]. Relative to the reference LoTP dataset, a substantial number of residues had low CC for either HiT or for HiP (Supplementary Fig. 11), suggesting differences in side-chain conformational ensembles due to these perturbations. For example, 19 (6.7%) residues had CC < 0.5 in HiT, and 22 (7.8%) residues had CC < 0.5 in HiP. Excluding the flexible E loop (residues 375–383), 18 (6.6%) residues had CC < 0.5 in HiT, and 14 (5.1%) residues had CC < 0.5 in HiP. If high temperature vs. high pressure had similar structural effects, a similar set of residues would be expected to have low CC for both HiT and HiP (each relative to LoTP). However, relatively few residues fall into this category (purple bars in Supplementary Fig. 11), suggesting that temperature vs. pressure are complementary perturbations that affect different areas of the protein.

To validate these quantitative Ringer results, we examined the models and density maps in detail for several examples. For most residues, the Ringer curves are indeed similar across all three datasets (Fig. 5a). For other residues, by contrast, the curves differ in one or more datasets, indicating perturbation-induced changes to side-chain conformations. For example, Glu519 adopts the same χ1 rotamer for LoTP and HiP, but a different χ1 rotamer at HiT (Fig. 5b; see also Fig. 3c), involving *a* ~ 0.3 Å backbone shift (Fig. 3a). By contrast, Ser327 adopts different primary χ1 rotamers for LoTP, HiT, and HiP (Fig. 5c). Some residues had distinct Ringer curves at HiP relative to LoTP and HiT (Supplementary Fig. 12) but were associated with distinct

backbone positions of the E loop that occurred only at HiP (Figs. 3a and 4).

As the Ringer curves above only account for the first side-chain torsion angle (χ1), we also compared rotamer names, which account for all side-chain torsion angles[50] (see "Methods" section). Excluding the flexible E loop, relatively few residues had different rotamers as alternate conformations in the same model: 7 for LoTP, 7 for HiT, and 7 for HiP. However, compared to LoTP, 50 residues (18%) had a different rotamer in HiT, and 38 residues (14%) had a different rotamer in HiP. Thus, by contrast to only the side-chain base as measured by Ringer, temperature, and pressure both have greater effects on the overall conformations of side chains, stabilizing distinct energy basins. Moreover, of the residues that differed from LoTP, 29 were unique to either only HiT or only HiP, indicating distinct conformational effects from temperature vs. pressure.

Finally, beyond just torsion angles for individual side chains, we explored whether many torsion angles distributed throughout the protein structure may undergo correlated changes in response to temperature vs. pressure. A recent tool called RoPE showed that linear combinations of backbone and side-chain torsion angles in a reduced-dimensionality space can help reveal new insights into the key differences between sets of structural models[42]. Using RoPE analysis, we examined our three new structures relative to all previous STEP structures (Fig. 6), leading to several interesting observations.

First, the STEP structures generally cluster based on resolution, as noted previously for other proteins[42], with our new structures at intermediate-to-high resolution compared to prior structures. Second, each set of structures with a consistent crystal form clusters together: (i) our new structures plus 2bv5, (ii) most remaining structures, and (iii) the mouse STEP structure 6h8s. Third, most of the previous structures segregate into an active-like cluster (either allosterically activated or bound to a substrate peptide) or an inactive-like cluster (bound to orthosteric inhibitors), indicating that subtle signatures of the protein's functional state are embedded in torsion-angle space. The allosterically activated structure (6h8s) is nearest to other active-like structures, despite it being mouse-derived (91% sequence identity to human STEP) and having a unique crystal

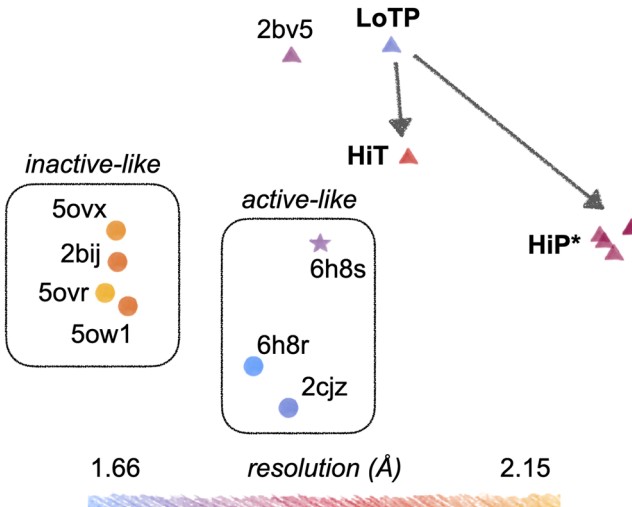

**Fig. 6 Dimensionality reduction in torsion-angle space reveals clustering based on several factors including temperature vs. pressure.** Our new structures (LoTP, HiT, HiP) are shown relative to all previous STEP structures from the PDB in RoPE reduced-dimensionality torsion-angle space[42]. Horizontal and vertical axes correspond to different combinations of the top principal component analysis (PCA) modes. Apparent inactive-like vs. active-like clusters are highlighted. Different icon shapes indicate distinct crystal forms (unit cell parameters). Resolution is shown by color. Arrows indicate the effects of high temperature vs. pressure relative to our reference structure. HiP* indicates several HiP models with the E loop prepared in different ways; see Methods. 6h8s: mouse STEP; all other structures: human STEP. All models were prepared for RoPE with PDB-REDO[72] to ensure consistent treatment[42].

form; hence, signatures of the protein's inherent functional state appear to persist in this space despite differences in amino acid sequence and crystal lattice.

Fourth, whereas our LoTP structure is near the most analogous previous structure (2bv5) in torsion-angle space as expected, our HiT and HiP structures move in distinct directions from this reference point (Fig. 6). Notably, HiT moves toward the active-like structures, whereas HiP moves away from all known STEP structures. This is despite only HiP featuring a conformation of the E loop resembling the allosterically activated structure 6h8r (Fig. 4 and Supplementary Fig. 8), but consistent with only HiT featuring side-chain and backbone conformations of the active-site Q loop in a putatively active-like state (Figs. 3c and 5b). HiP models with significantly different E-loop conformations and prepared for analysis in different ways have similar positions in torsion-angle space, confirming that RoPE analysis highlights structurally distributed as opposed to localized features. Relative to LoTP, the coordinated torsion-angle changes in HiT and HiP detected by RoPE visually correspond to hinging of the first half of the primary structure (initial α-helices + loops) relative to the second half (β-sheet + α-helical bundle), albeit with apparently meaningful differences between them given their large separation in RoPE space. Overall, these results indicate that although high pressure induces an active-like conformation locally in the E loop, high temperature induces a more global, distributed active-like state of the protein.

## Discussion

While temperature is growing in use as an experimental perturbation in macromolecular crystallography, pressure has received less attention for such applications. Here we show that both

temperature and pressure enact distinct and significant effects on the conformational ensemble of STEP, not only globally but also locally at several key functional areas.

In our structures, high temperature increases both unit cell volume and protein molecular volume (Table 2) as seen previously[9]. High pressure decreases unit cell volume as seen previously[27,51,52], yet still slightly increases protein molecular volume (Table 2). Thus, intriguingly, when subjected to pressure, the STEP protein molecule itself slightly expands, even as its environment is compressed. This differs from previous high-pressure crystal structures of other proteins with a decreased protein molecular volume[27,29,31], but agrees with a high-pressure NMR solution structure with a slightly increased protein molecular volume[53]. The slight increase in molecular volume of the protein itself that we observe upon pressurization may be initially unintuitive, but can likely be explained by counterbalancing decreases in the volume of the bulk solvent (which is invisible to crystallography) within the crystal lattice, and is thus consistent with the thermodynamic expectation that pressurization decreases the molar volume of the protein-solvent system. Nonetheless, our observations suggest that the mechanisms by which pressure impacts the detailed conformational landscapes of different proteins are complex and potentially context-sensitive.

Although many of the structural changes we see could be considered small, it is important to remember that sub-angstrom shifts can be directly relevant to protein function[26]. This is consistent with our RoPE results, in which HiT vs. HiP have very distinct characteristics despite the overall structures being apparently similar. Crucially, the difference in the E loop at HiP does not dominate the signal (see Fig. 6 and Methods), indicating that the differences between high temperature vs. pressure are driven by smaller, subtler conformational changes distributed throughout the tertiary structure.

The largest conformational changes we observe in STEP are in the E loop (Fig. 4a), a conserved loop in PTPs that plays a critical role in regulation[54]. Only at HiP do we see evidence in the electron density for a dual-conformation E loop (Fig. 4b–d). Both conformations were individually evident in previous structures of STEP with different chemical modifications or allosteric ligands (Supplementary Fig. 8). Our data indicate that applying a physical perturbation (pressure) is sufficient to induce these conformations to coexist in a single crystal of the unliganded protein, which has implications for accessing excited states of other proteins.

Beyond the E loop, we observe new conformations not captured in previous structures of STEP. For instance, only at HiT, we see Glu519 of the active-site Q loop adopt a new side-chain rotamer that engages in an interaction with Lys439. Notably, Lys439 follows the WPD sequence, forming a WPDQK sequence. Recently, a conserved PDFG motif was proposed to underlie the ability of the WPD loop to toggle between discrete open vs. closed states in PTPs[55]. However, the corresponding residues in STEP are PDQK — and, as revealed by our HiT structure, the final K (Lys439) engages with the Q loop nearby. Perhaps not coincidentally given these idiosyncratic features, the WPD loop of STEP has not been observed in the usual open or closed states as with most other PTPs, but only in the atypically open state[39,41]. Together, these observations suggest that the STEP active site does not adhere to expectations from the rest of the PTP family, and points to specific amino acids and conformations that may encode its unusual behavior. We speculate that these unique structural properties of STEP likely underlie its substantially lower catalytic activity relative to other PTPs like PTP1B[11,40,56] and may be related to its unique physiological roles in neuronal development[57].

It is plausible that this unresponsive, atypically open state could be modulated by binding of regulators or alterations in the cellular environment, creating a way to regulate catalysis. In this light, we observe two sulfates simultaneously bound within the active site. The biological significance of this observation for STEP is not immediately clear. It is likely that binding of sulfate-like moieties in the top site, as in several orthosteric inhibitors (Supplementary Fig. 2b), blocks closure of the WPD loop, effectively wedging it atypically open (Supplementary Fig. 2e). However, even with the top site free and substrate bound only to the bottom site, the WPD loop still remains atypically open in crystals[41] (Supplementary Fig. 2c). Removing all tightly bound molecules from the active site of STEP in future crystallographic studies could provide more definitive answers about the conformational landscape of this functionally critical but unusual catalytic loop.

Previously, based on computational simulations, a small-molecule allosteric activator for STEP was reported to enact its effects via a pair of allosteric pipelines[40]. We do not observe obvious shifts along these pathways at high temperature or pressure. However, we do observe shifts in the activator binding pocket itself (Supplementary Fig. 9). Although subtle, these conformational shifts may be sufficient to influence ligand-binding energetics, and therefore may aid structure-based drug design efforts to improve upon the relatively weak reported activator.

Beyond the reported allosteric activator site, we also observe perturbation-sensitive shifts at other known or putative ligand binding sites. First, at LoTP and HiP, an ordered glycerol molecule is bound near α2′ and the Q loop. By contrast, at HiT, ordered waters are present instead, and α1′-α2′ and the Q loop undergo conformational shifts (Fig. 3c). Importantly, all three structures are from crystals treated with similar glycerol-containing cryoprotectant solutions. It is thus plausible that crystal cryocooling induces the glycerol to bind[14], preventing nearby conformations with potential functional relevance seen at physiological temperature (Fig. 3c and 6). Second, in the paralogous PTP SHP2, the α1′-α2′ region helps form the binding site for the potent allosteric inhibitor SHP099, although the mechanism also involves additional domains[58]. The corresponding α1′-α2′ region in STEP is not known to be allosteric. However, the subtle but coordinated conformational changes we observe here at physiological temperature raise the enticing possibility that some aspects of the allosteric capacity demonstrated in SHP2 are also present in STEP, and perhaps even other PTPs.

In general, allostery in STEP remains poorly understood, hindering efforts to elucidate this important protein's endogenous regulatory mechanisms and to develop specific allosteric modulators. To address this important gap, several approaches should be considered. First, exploiting different crystal forms, including those in the allosteric-activator-bound structures for human and mouse STEP[40], may provide new windows into conformational mobility otherwise masked by crystal contacts. Second, higher pressures than those reported here[32] may enable access to additional excited states. Third, X-ray diffraction at high pressure and physiological temperature simultaneously has the potential to reveal unique aspects of conformational landscapes not evident from a single perturbation alone. More broadly, the avant-garde crystallographic and computational methods outlined here should prove useful tools to investigate allosteric mechanisms in a variety of other proteins, including but not limited to other PTP family members that also exhibit an atypically open WPD loop such as LYP[59].

Overall, the work reported here is consistent with the notion that proteins sample conformations from a multifaceted energy landscape, and that different physical perturbations such as temperature and pressure can access distinct, complementary features of this landscape, thus opening doors to elucidating fundamental connections between protein structural dynamics and function.

## Methods

**Molecular biology**. A plasmid containing the catalytic domain [258–539] of STEP (PTPN5) with an N-terminal 6xHis & TEV cleavage site was obtained via Addgene from Nicola Burgess-Brown (Addgene plasmid #39166; http://n2t.net/addgene:39166; RRID:Addgene_39166). This was transformed into BL21(DE3) Rosetta2 (pRARE2) cells (MilliporeSigma). The sequence of the insert was independently verified using Sanger sequencing, with standard T7 promoter primers.

**Protein expression**. In all steps, the antibiotics chloramphenicol (Cam) and ampicillin (Amp) were used to maintain selection at working concentrations of 30 μg/mL and 100 μg/mL respectively. Previously transformed cells from glycerol stocks were plated on an LB-Agar + Amp + Cam plate and incubated overnight at 37 °C. Individual colonies were picked and grown up overnight at 18 °C in LB + Amp + Cam starter cultures (10 mL), shaking at 180 rpm. This starter culture was then added into baffled flasks containing 1 L of LB+Amp+Cam media, and incubated to OD 0.6–0.8 at 37 °C, with shaking at 180 rpm. Expression was then induced by adding IPTG to a final concentration of 0.2 mM; cultures were then incubated overnight at 18 °C, shaking at 180 rpm, before cells were harvested by centrifugation at 3000 rpm for 45 min, snap frozen in liquid $N_2$ and stored at -80 °C.

**Protein purification**. Frozen cellets (cell pellets) were thawed on ice, then 30 mL lysis buffer (50 mM HEPES pH 7.5, 500 mM NaCl, 5 mM imidazole, 5% v/v glycerol, 2 mM DTT) was added. One Pierce EDTA-free protease inhibitor mini-tablet per cellet was also added, and resuspended in a vortexer. Cells in the slurry were then lysed by 3 passages through a cell homogenizer (Avestin) operating with 1000 bar peak. Lysate was then centrifuged for 45 min at 50,000×g to spin down the cell fragments. The supernatant was filtered through a 0.22 μm filter to remove final cell debris.

A 5 mL Ni-NTA column (Cytiva) was equilibrated in freshly prepared low-imidazole buffer (50 mM HEPES pH 7.5, 500 mM NaCl, 30 mM imidazole, 5% v/v glycerol, 2 mM DTT). The lysate supernatant was applied to this column, washed with 2 column volumes (CV) of low-imidazole buffer, then gradient-eluted over 10 CV to 100% high-imidazole buffer (50 mM HEPES pH 7.5, 500 mM NaCl, 500 mM imidazole, 5% v/v glycerol, 2 mM DTT), collecting in 5 mL fractions. The STEP-containing fractions eluted around the 40% gradient mark were collected, concentrated using a 15 mL Centriprep 10 K spin-concentrator (Millipore) to a final volume of 5 mL, and filtered through a syringe-mount 0.22 μm filter to remove unidentified precipitate.

A Sephadex 20/200 column (Cytiva) was equilibrated with 2 CV of SEC buffer (50 mM HEPES pH 7.5, 500 mM NaCl, 5% v/v glycerol, 2 mM DTT). The concentrated, filtered Ni-binding fraction was injected onto a 5 mL loop, loaded onto the column, and fractionated over 2 CV, collecting 1 mL fractions. Two peaks were observed, and the fractions corresponding to the largest, STEP-containing peak were pooled.

A HiTrap Q HP anion-exchange column (Cytiva) was equilibrated with 2 CV of low-salt buffer (50 mM HEPES pH 7.5, 10 mM DTT). The pooled peak from size-exclusion chromatography was diluted to a final volume of 100 mL by

addition of low-salt buffer, and filtered through a 0.22 μm bottle-top vacuum filter (Celltreat). This was then applied to the Q column, washed with 2 CV of low-salt buffer, and then gradient-eluted over 5 CV to 100% high-salt buffer (50 mM HEPES pH 7.5, 1000 mM NaCl, 10 mM DTT) collecting 5 mL fractions. A single STEP-containing peak was collected at 40% gradient mark.

This final STEP protein was concentrated in Centriprep 10 K spin-concentrators to 3 mL volume, and then further concentrated in Amicon 10 K spin-concentrators to a final concentration of 10 mg/mL, as measured by Nanodrop, and used fresh as the protein sample in crystallography. The identity of STEP vs. other proteins/contaminants was confirmed using SDS-PAGE gels at each step of the purification.

**Crystallization and crystal preparation.** Precipitant well solution (30% PEG 3350, 200 mM $Li_2SO_4$, 100 mM bis-tris pH 5.65) was prepared fresh. A Mosquito (SPT Labtech) was used to prepare 96-well 3-drop Intelliplate Low-profile (Art Robbins Instruments) plates. 80 μL well solution was placed into the reservoir. Three 1 μL drops at a protein concentration of 10 mg/mL were placed per well, using 2:1, 1:1, and 1:2 ratios of well solution to protein sample. Crystallization drops were incubated at room temperature. Crystals nucleated within 3 days, mostly in 1:1 droplets, and grew over a week to around $80 \times 80 \times 20$ μm.

For the ambient-pressure low-temperature 100 K (LoTP) dataset, the crystal was soaked in cryoprotectant (mother liquor + 15% v/v glycerol), and cryocooled with liquid nitrogen.

For the high-pressure (205 MPa) cryogenic-temperature dataset (HiP), the crystal was looped in a 100 μm loop, soaked in cryoprotectant (mother liquor + 15% v/v glycerol), and placed in a capillary with cryoprotectant at the end of the tube for shipping to CHESS. At CHESS, the capillary was removed, and the crystal was coated in NVH oil. The crystal was pressurized for 20 min at 205 MPa, cryocooled with liquid nitrogen under pressure, and stored under liquid nitrogen thereafter.

High-temperature diffraction required larger crystals, and so were prepared in Nextal EasyXtal 15-well hanging-drop trays. Precipitant well solution was prepared with the same composition as above (pH 5.5). In all, 400 μL well solution was placed into the reservoirs. Three 3 μL drops at a protein concentration of 10 mg/mL were placed per well, using 1:1 ratios of well solution to protein sample each. Crystallization drops were incubated at room temperature. Crystals nucleated within 3 days, and grew over a week to around $140 \times 70 \times 40$ μm.

For the high-temperature ambient-pressure 310 K (HiT) dataset, the crystal was soaked in cryoprotectant (mother liquor + 15% v/v glycerol), coated in NVH oil, and placed in a capillary with cryoprotectant at the end of the tube for shipping to CHESS.

**X-ray data collection.** All X-ray diffraction datasets were collected at the ID7B2 (FlexX) beamline for macromolecular X-ray science at the Cornell High Energy Synchrotron Source (Mac-CHESS), Ithaca, New York, USA, using an X-ray beam energy of 12 keV and corresponding wavelength of 1.033 Å. The LoTP dataset was collected using beam dimensions of $30 \times 20$ μm, flux of $5 \times 10^{11}$ ph/s, rotation rate of 2°/s, and no translation. The HiT dataset was collected using beam dimensions of $30 \times 20$ μm, flux flux of $1.6 \times 10^{10}$ ph/s, rotation rate of 10°/s, and translation (i.e. helical/vector data collection) along the length of the approximately $140 \times 70 \times 40$ μm crystal. The HiP dataset was collected using beam dimensions of $100 \times 100$ μm, flux of $2 \times 10^{10}$ ph/s, rotation rate of 1°/s, and no translation.

**X-ray data reduction and modeling.** Data reduction and modeling was performed similarly for all three datasets, with the data reduction pipeline DIALS[60]. The LoTP dataset was trimmed to the first 130 (out of 180) frames due to increased ice inclusions in later frames. Resolution cutoffs were determined automatically by DIALS based on a combination of $CC_{1/2}$, I/sigma(I), $R_{merge}$, and completeness[60]. Molecular replacement was performed via Dimple[61], with subsequent refinement performed using REFMAC[62] and phenix.refine[63], with models manually adjusted between rounds of refinement using COOT[64]. Hydrogens were added using phenix.ready_set[65]. X-ray/stereochemistry weight, X-ray ADP weight, and occupancies were all refined and optimized during the final rounds of refinement. Model validation statistics were generated using MolProbity[66]. Solvent content was calculated via MATTPROB[67,68]. Ramachandran outliers (%) are 1.07, 0.36, and 0.71 for LoTP, HiT, and HiP, while Ramachandran favored (%) values are 95.00, 95.00, and 93.21, respectively. Data collection and refinement statistics can be found in Table 1.

To complement/validate the conclusions from the manually modeled waters in the deposited models, automated analysis of structural waters was performed by truncating our datasets to the same resolution (1.96 Å) in the PHENIX GUI reflection file editor. After removing all manually modeled heteroatoms including waters, our models were subjected to refinement with default settings, followed by both Cartesian and torsion angle simulated annealing refinement with a start temperature of 4000 K. Waters were automatically added based on real space density using the Coot "find waters" tool. Finally, the models were refined with phenix.refine, first without then with automatic water updating.

Polder maps and omit maps were calculated using the phenix.polder utility from the PHENIX GUI[45].

**Model analysis.** Cα distances between structures were calculated using VMD[69]. Rotamer names were calculated using phenix.rotalyze[66] based on the latest rotamer distributions from MolProbity[70].

Protein volumes were calculated using the ProteinVolume software[71]. Values for the "total volume" output were nearly identical whether waters were included or not, and were similar (conclusions did not change) when state A vs. state B of the HiP structure were analyzed.

Ringer[4] was run on models that only contained a single conformation for each residue, with alternate conformations removed using phenix.pdbtools. For HiP, each E-loop conformation was treated individually. The single-conformer models each underwent multiple cycles of refinement using phenix.refine. The refined models and maps were then used as input to Ringer. Plots were generated using *ggplot2* and *ggbreak*.

For RoPE analysis, all structures were pre-processed with PDB-REDO[72] to ensure consistent treatment, as recommended[42]. The different HiP points in Fig. 6 correspond to different preparations of the model: run PDB-REDO for deposited model; extract each state, run PDB-REDO, then set all occupancies to unity; or extract each state, set all occupancies to unity, then run PDB-REDO.

**Statistics and reproducibility.** Calculation of correlation coefficients for Ringer analysis in Fig. 5 and Supplementary Figs. 11 and 12 was performed using the Pearson correlation method and Supplementary Data 2, and Supplementary Fig. 4 was performed using Supplementary Data 3. The analysis of the effects on pockets and cavities in Supplementary Fig. 10 was performed using a Welch's two-sample *t*-test with the pocket information in Supplementary Data 4. Two comparisons were made: one between HiP and LoTP, and the other between HiT and LoTP.

**Reporting summary**. Further information on research design is available in the Nature Portfolio Reporting Summary linked to this article.

## Data availability

Coordinates and structure factors that were generated during the course of this study have been deposited in the Protein Data Bank with the accession codes 8sls (STEP at cryogenic temperature and ambient pressure), 8slt (STEP at physiological temperature and ambient pressure), and 8slu (STEP at cryogenic temperature and high pressure). The protein structure used as a search model for molecular replacement is accessible in the Protein Data Bank under accession codes 2bv5.

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

## Acknowledgements

DAK is supported by NIH R35 GM133769. We thank Helen Ginn for help with RoPE analysis, members of the CUNY Advanced Science Research Center (ASRC) Structural Biology Initiative (SBI) for helpful discussions, Akshay Raju and Shivani Sharma for help with PTP bioinformatics, and Marian Szebenyi for help with arranging our X-ray beamtime. This work is based upon research conducted at the Center for High Energy X-ray Sciences (CHEXS), which is supported by the National Science Foundation under award DMR-1829070, and the Macromolecular Diffraction at CHESS (MacCHESS) facility, which is supported by award 1-P30-GM124166-01A1 from the National Institute of General Medical Sciences, National Institutes of Health, and by New York State's Empire State Development Corporation (NYSTAR).

## Author contributions

L.G. analyzed data and wrote the manuscript. A.E. analyzed data and wrote the manuscript. B.T.R. designed experiments, performed experiments, and edited the manuscript. M.K. designed experiments and performed experiments. Q.H. designed experiments and edited the manuscript. A.D.F. performed experiments and edited the manuscript. D.A.K. conceived the study, designed experiments, analyzed data, and wrote the manuscript.

## Competing interests

The authors declare no competing interests.
