## [Peer Review File · Communications Biology]

Reviewers' comments:

Reviewer #1 (Remarks to the Author):

The manuscript provides further evidence that single crystal X-ray diffraction can expose low-populated states in proteins when subjecting the samples to different environmental conditions other than the now standard cryogenic data collection. The article focuses on the clinically relevant protein tyrosine phosphatase STEP (PTPN5) and interestingly describes new conformations at physiological temperature (310 K) or high pressure (205 MPa). Intriguingly, the substates reported may point to a new allosteric hotspot that might be shared among PTPs.

Major comments

A. The presence of sulfate groups in the active site points to an analog of the phosphoenzyme intermediate rather than a ligand-free structure (the apo term should be used only for enzymes missing a prosthetic group). A superimposition of the PTP1B phosphoenzyme intermediate (1A5Y) with LoTP (8SLS) and HiT (8SLT) explains positive and negative electron density peaks around the catalytic Cys472, therefore a phosphoenzyme-like substate should be included in those structures. Polder maps of Cys472 reveal both substates in all structures but at different populations, which could be interesting when interpreting the results. Finally, the Ringer output for Cys472 should also reflect this.

B. The authors should clarify early in the text what makes a STEP structure active-like or inactive-like.

C. The authors claim changes in the ordered solvent by a temperature or pressure effect. This analysis must consider that the number of detectable water molecules changes with resolution. One way would be to truncate all datasets to the lowest resolution (1.96 Å HiT). In addition, a consistent water placement method should be used. Inspection of water molecules in the reported crystal structures reveals inaccuracies such as the lack of signal or flooding of unknown electron density blobs (e.g., waters 706, 735, 823, 854 in 8SLS).

Minor comments

1. Line 92/93. It is unclear why kinases were included considering no kinase was under investigation in this study.
2. Line 125. The claim "a new side-chain rotamer" contradicts the latter "previously was only seen" in line 127.
3. Line 446-448. To strengthen the argument that torsion-angle analysis can cluster active from inactive structures (with low sensitivity to sequence identity), faster enzymes like PTP1B should cluster around the active forms of STEP.
4. Line 476-479. The suggestion of a conserved allosteric hotspot across different PTPs is exciting!
5. Line 590. Given that the resolution cutoff was based on CC1/2, why is HiP CC1/2 (0.68) higher than the ones used for LoTP (0.374) and HiT (0.358)?
6. Line 617. The 8SLT is cited twice; the last one should be 8SLU.

Reviewer #2 (Remarks to the Author):

This is a very interesting report contrasting structures of an enzyme obtained under contrasting

thermodynamics conditions of temperature and pressure. The authors are correct to note that although the observed changes in structure are small, such small changes can have significant effects on catalysis. This reviewer found the RoPE analysis pretty convincing. Particularly interesting are the results obtained at loTHiP, conditions under which very few studies have been reported, despite solution studies revealing that pressure allows access to excited states of proteins and other biomolecules. Thus, these results are timely. In addition, as also noted by the authors, much insight remains to be gained from high temperature crystallography, and the present results support that notion.

I have one point of disagreement with the interpretation of the results. The authors state: "... intriguingly, when subjected to pressure, the STEP protein molecule itself slightly expands, even as its environment is compressed. This differs from previous high-pressure crystal structures of 414 other proteins with a decreased protein molecular volume 27,29,31, but agrees with a high-pressure NMR solution structure with a slightly increased protein molecular volume, corresponding to negative compressibility 52. Our surprising, counterintuitive result suggests that the mechanisms by which pressure impacts the conformational landscapes of different proteins are complex and potentially context-sensitive."

Actually pressure effects on conformational equilibria are due to differences in molar, not molecular volume. In many cases, pressure leads to more open, larger (in terms of molecular volume) forms of proteins, but the volume of the protein-solvent system (the molar volume) in these cases is nonetheless lower. Because in crystallographic studies, not all hydrating water molecules are detected, it is not possible to estimate molar volume differences. Thus, in my view the observations are neither surprising nor counter-intuitive and need no negative compressibility to be explained. This said, more hydrated forms of proteins typically have lower compressibilities, so this is not contradictory. In any case, this interpretation should be revised.

Along these lines, the convincing differences between pressure and temperature effects are consistent with differences in thermal expansion coefficients and molar volumes generally observed between states. However, these detailed structural studies show that these two thermodynamic perturbations lead to changes in distinct protein regions. This intricate insight is extremely interesting. Typically, large local compressibilities or changes in local conformation can be linked to the existence of nearby cavities which either compress, are filled with solvent or are eliminated by the conformational change induced by pressure. In contrast, internal void volume tends to increase with increasing temperature. It would have been nice to see a comparative analysis of the localization, size and shape of internal cavities in these three structures.

Response to Reviewers

Pushed to extremes: distinct effects of high temperature vs. pressure on the structure of an atypical phosphatase

We thank the reviewers for their thoughtful comments. The resulting revisions have improved the manuscript. Below, reviewers' comments are in black text, our responses are in blue text, and modifications to the manuscript are in red text.

Reviewer #1 (Remarks to the Author):

The manuscript provides further evidence that single crystal X-ray diffraction can expose low-populated states in proteins when subjecting the samples to different environmental conditions other than the now standard cryogenic data collection. The article focuses on the clinically relevant protein tyrosine phosphatase STEP (PTPN5) and interestingly describes new conformations at physiological temperature (310 K) or high pressure (205 MPa). Intriguingly, the substates reported may point to a new allosteric hotspot that might be shared among PTPs.

We thank the reviewer for this summary of our work.

Major comments

A. The presence of sulfate groups in the active site points to an analog of the phosphoenzyme intermediate rather than a ligand-free structure (the apo term should be used only for enzymes missing a prosthetic group). A superimposition of the PTP1B phosphoenzyme intermediate (1A5Y) with LoTP (8SLS) and HiT (8SLT) explains positive and negative electron density peaks around the catalytic Cys472, therefore a phosphoenzyme-like substate should be included in those structures. Polder maps of Cys472 reveal both substates in all structures but at different populations, which could be interesting when interpreting the results. Finally, the Ringer output for Cys472 should also reflect this.

We agree that the density in the active site is a bit complicated to interpret. In our experience, this is often the case with structures of other PTPs (such as PTP1B) as well.

We feel that a phosphoenzyme intermediate is an unlikely explanation for our datasets. We attempted to model such an intermediate, but the result has strong negative difference peaks along the bond between the Cys472 C \$\beta\$ -S \$\gamma\$ bond (new Supp. Fig. 3). Moreover, the structure of PTP1B with a phosphoenzyme intermediate that the reviewer mentioned (PDB ID 1a5y) used a nearby Q262A mutation to stabilize the otherwise transient intermediate. Similarly, a previous structure of STEP with a bound phosphotyrosine, mimicking the Michaelis complex (PDB ID 2cjz), used a C472S mutation to stabilize the intermediate. By contrast, our STEP structures are WT, so there is nothing in place to stabilize a putative short-lived phosphoenzyme intermediate chemical state. Moreover, we did not

add phosphotyrosine or any similar molecule to our solutions, and used pure samples of recombinant human STEP purified from bacteria in our experiments.

However, we feel that the reviewer's attention to this area of our structures was warranted, as upon further inspection we do feel that Cys472 in our structures could arguably be better modeled as exhibiting dual conformations for LoTP and HiT, yet a single conformation for HiP -- consistent with the reviewer's general observation that Cys472 may sample different substates but in different populations for our different datasets. This modeling choice is justified by Ringer analysis (see new Supp. Fig. 4) as well as Polder and omit maps (see new Supp. Fig. 5-6), analyses that the reviewer suggested.

To address these points, we have added the aforementioned new Supp. Figures 3-6 as well as the following Results text to the paper:

The "bottom" sulfate is well-coordinated by the catalytic P loop, analogous to the phosphate group in the phosphotyrosine (pTyr) substrate in PDB ID 2cjz⁴¹ (**Supp. Fig. 2c**). A phosphocysteine reaction intermediate with a covalent bond to Cys472 is an unlikely explanation of our data, as refining such a putative model resulted in strong negative difference density peaks (**Supp. Fig. 3**), our crystals contained high concentrations of lithium sulfate but not any phosphate-containing compounds, and previous intermediate-bound PTP structures used inactivating mutations to capture such intermediates^{41,45} whereas our structure is wildtype. In our structures, as supported by strong electron density, Cys472 predominantly adopts a side-chain rotamer that points away from the sulfate, thus avoiding a steric clash. This rotamer in our new structures is rare for STEP: it was previously only seen as a partial-occupancy alternate conformation in an allosterically activated structure (PDB ID 6h8r) (**Supp. Fig. 2b**). Further supporting this primary rotamer, Ringer curves⁴ for Cys472 for all three datasets have a dominant peak for the χ_1 side-chain dihedral angle near 180°. In addition, Ringer curves from different model preparations for all three datasets also have a secondary χ_1 peak near +60°, suggesting sensitivity of Ringer to precise input coordinates and/or subtle sensitivity to our global perturbations (**Supp. Fig. 4**). This secondary Ringer peak is consistent with an alternate rotamer conformation for Cys472, which is further supported by unbiased Polder⁴⁶ (**Supp. Fig. 5**) and omit (**Supp. Fig. 6**) electron density maps. Moreover, the backbone density for Cys472 is consistent with multiple positions, suggesting C α displacements that are perpendicular to the chain direction (**Supp. Fig. 5, Supp. Fig. 6**) and similar in magnitude (0.44–0.93 Å) to those seen between alternate conformations in a previous structure of STEP (0.73 Å for PDB ID 2bv5, although Cys472 is acetylated in that structure and does not change rotamer). Taken together, these observations support the interpretation that Cys472 samples both a rare primary rotamer and a low-occupancy alternate rotamer that is sterically mutually exclusive with a fortuitously observed sulfate bound at high but non-unity occupancy.

Supplementary Figure 3: Phosphocysteine is a poor fit for the catalytic Cys472.

Refinement of models with a putative phosphocysteine intermediate covalently bound to the catalytic Cys472 result in reasonable fit to parts of the 2Fo-Fc (blue mesh, 1.0 σ) electron density, but unacceptable fits to the Fo-Fc difference electron density (green mesh, +3.0 σ ; red mesh, -3.0 σ).

- a) LoTP.
- b) HiP.
- c) HiT.

Supplementary Figure 4: Ringer analysis of the catalytic cysteine.

- Ringer curve for Cys472 using input model prepared by extracting alternate conformation A (primary rotamer, pointed away from sulfate) from the dual-conformation model and performing reciprocal-space refinement in PHENIX.
- Ringer curve for Cys472 using input model prepared by extracting alternate conformation A from the dual-conformation model, truncating to C β , performing real-space refinement in Coot, and restoring the rest of the side chain with the standard library rotamer closest to the original alternate conformation A in Coot, with no further refinement. This strategy aims to bypass the constraints on backbone positioning imposed by particular side-chain rotamers during refinement.
- Models from a–b), demonstrating differences in C α -C β vectors, which can influence Ringer results. *Left*: LoTP (blue), *middle*: HiT (red), *right*: HiP (green). Shown in each panel are alternate conformation A from a) (darkest color), adjusted input conformation from b) (intermediate color), and alternate conformation B for visual reference (lightest color).

Supplementary Figure 5: Polder maps for the catalytic cysteine.

Left: Our dual-conformation model of Cys472 is supported by a Polder map (5σ), including the backbone C α atom (arrow).

Right: A putative single-conformation model for Cys472, obtained by extracting only the primary conformation from our dual-conformation model, highlights Polder density consistent with a missing secondary conformation, particularly noticeable for the backbone including the C α atom (arrow).

- a) LoTP.
- b) HiT.
- c) HiP.

Supplementary Figure 6: Omit maps for the catalytic cysteine.

Same as **Supp. Fig. 5**, but using omit maps (3σ) instead of Polder maps.

In light of the reviewer's suggestion about the usage of "apo", we have also changed several instances to "unliganded" to emphasize that in our structures the enzyme has not been exposed to a physiological ligand or substrate nor is it undergoing catalysis.

In addition, because our structural models changed (slightly), we regenerated all figures in the paper that could possibly be impacted by this change; this resulted in no changes to our conclusions.

B. The authors should clarify early in the text what makes a STEP structure active-like or inactive-like.

We thank the review for their comments and agree that we should clarify earlier in the text what makes the structures active- or inactive-like. We have therefore added the following text:

The public Protein Data Bank ³⁷ includes 8 high-resolution (1.66–2.15 Å) crystal structures of STEP (7 human, 1 mouse) with different ligands, demonstrating its tractability with crystallography. Of these structures, 3 are in an "inactive-like" state, either bound to a competitive inhibitor ³⁸ or inactivated through the acetylation of the catalytic cysteine ³⁹, while another 3 are in an "active-like" state, either bound to an allosteric small-molecule activator ⁴⁰ or in a Michaelis-like complex with a pTyr substrate bound to a catalytic C472S mutant ⁴¹.

C. The authors claim changes in the ordered solvent by a temperature or pressure effect. This analysis must consider that the number of detectable water molecules changes with resolution. One way would be to truncate all datasets to the lowest resolution (1.96 Å HiT). In addition, a consistent water placement method should be used. Inspection of water molecules in the reported crystal structures reveals inaccuracies such as the lack of signal or flooding of unknown electron density blobs (e.g., waters 706, 735, 823, 854 in 8SLS).

We appreciate the reviewer's useful comments towards detecting waters given the differences in resolution between our datasets. Upon further inspection, we agree that some of our waters were over-modeled. We have revisited our models, removed some waters, and re-refined. Of the waters the reviewer mentioned as examples, water 706 in 8SLS has a nice H-bond to nearby Glu366 and has nice 2Fo-Fc density at 1.2 sigma, so we preserved it; however, other waters the reviewer mentions were indeed questionable, so we deleted them. The updated refinement statistics are just as good as before (see updated Table 1). We have re-deposited these updated models to the PDB using the same PDB codes via the PDB versioning system. With these new models, the overall trend still holds that LoTP has the most waters, HiT has the fewest, and HiP is in between (see new Supp. Table 1). In addition, a similar number of waters are unique to each dataset (see updated Fig. 2). As noted above, because our structural models changed, we regenerated all figures in the paper that could possibly be impacted by this change; this resulted in no changes to our conclusions.

To validate this manual modeling, and to more directly address the reviewer's point about differences in resolution, we have also truncated the datasets to a common resolution as the reviewer suggests, and used automated water placement to avoid any potential bias in modeling. Our strategy was intended to put all three datasets on equal footing and eliminate any potential of bias or inconsistency from manual water placement, at the possible expense of missing some small number of waters due to the conservative nature of the Coot water placement algorithm (although by subsequent visual inspection such missed waters were quite few). We have added the following new Methods text to describe our protocol:

To complement/validate the conclusions from the manually modeled waters in the deposited models, automated analysis of structural waters was performed by truncating our datasets to the same resolution (1.96 Å) in the PHENIX GUI reflection file editor. After removing all manually modeled heteroatoms including waters, our models were subjected to refinement with default settings, followed by both Cartesian and torsion angle simulated annealing

refinement with a start temperature of 4000 K. Waters were automatically added based on real space density using the Coot “find waters” tool. Finally, the models were refined with phenix.refine, first without then with automatic water updating.

Compared to our newly revised manual models (see above), this automated procedure results in quite similar numbers of waters for all three datasets (see new Supp. Table 1). With these new alternative, automated models, the overall trend still holds that LoTP has by far the most waters, HiT has by far the fewest, and HiP is in between. In fact, with these revisions, the numbers of unique waters increased relative to the original version of this manuscript (e.g. total number of unique waters increased from 84 to 105).

To address these points, we have added the aforementioned new Supp. Table 1 as well as the following Results text (plus associated Methods text above) to the paper:

In addition to global changes to the crystal lattice, high temperature and pressure have striking effects on the solvation layer surrounding the protein. In our manually modeled, deposited structures, LoTP has by far the most waters, HiT has by far the fewest, and HiP has an intermediate number (**Supp. Table 1, Fig. 2**). To confirm that this result is not due to the small differences in resolution between datasets (**Table 1**), we truncated the LoTP and HiP diffraction data to match the resolution of the HiT dataset (1.96 Å), then performed fully automated, unbiased water placement for all three structures (see Methods). The resulting water counts are similar to the counts of manually placed waters in our deposited structures (**Supp. Table 1**), thus validating the conclusions derived from the latter.

Turning to specific water positions in our deposited structures, 17 (24.6%) of the HiT waters and 23 (23.5%) of the HiP waters were distinct from any LoTP water (> 2 Å, accounting for crystal symmetry) (**Fig. 2**). Of these 40 new positions, only 1 (2.5%) is common to both HiT and HiP. This suggests that high temperature and high pressure do not merely retain a subset of ordered waters, but rather stabilize new water positions, resulting in a distinct pattern of solvation. As shown below, some of these unique waters are located at functional sites in STEP (**Fig. 3**). In total, we reveal 67 (LoTP) + 16 (HiT) + 22 (HiP) = 105 waters that are unique to one structure (**Fig. 2**), further underscoring the value of crystallography with different axes of perturbations for mapping accessible patterns of protein solvation.

Figure 2: Ordered water molecules are sensitive to temperature and pressure.

(a-c) All ordered water molecules at (a) LoTP, (b) HiT, and (c) HiP are shown.

(d) Only the waters unique to each structure, i.e. > 2 Å from any water in the other two structures.

Coloring for active-site loops as in Fig. 1.

	LoTP	HiT	HiP
# waters, deposited v1 *	170	91	112
# waters, deposited v2	149	69	98
# waters, automated with truncated data	141	69	93

Supplementary Table 1: Waters are sensitive to perturbations but robust to modeling method.

See Methods for details of structure factor data truncation and automated water placement.

(* Not included in paper -- only for reviewer purposes)

Note that in the copy of Supp. Table 1 provided above we have included a row for original deposited models solely for the reviewer's convenience; this does not appear in the revised version of our paper, which pertains solely to our revised models and automated models.

Minor comments

1. Line 92/93. It is unclear why kinases were included considering no kinase was under investigation in this study.

We thank the reviewer for their valuable comments. We had included a reference to kinases in this paragraph as they work in opposite action to phosphatases, and are thus relevant to understanding the biological roles of phosphatases in cells. However, we agree that, as this is the only place in the manuscript that we mention kinases, it may be confusing for the audience. We have therefore removed the kinase remarks from this paragraph.

2. Line 125. The claim “a new side-chain rotamer” contradicts the latter “previously was only seen” in line 127.

We agree with the reviewer that this wording may be confusing to the reader. We have adjusted the wording to clarify that this rotamer is not new per se but is indeed rare for STEP:

In our structures, as supported by strong electron density, Cys472 predominantly adopts a side-chain rotamer that points away from the sulfate, thus avoiding a steric clash. This rotamer in our new structures is rare for STEP: it was previously only seen as a partial-occupancy alternate conformation in an allosterically activated structure (PDB ID 6h8r) (Supp. Fig. 2b). Further supporting this primary rotamer, Ringer curves⁴ for Cys472 for all three datasets have a dominant peak for the χ_1 side-chain dihedral angle near 180°.

3. Line 446-448. To strengthen the argument that torsion-angle analysis can cluster active from inactive structures (with low sensitivity to sequence identity), faster enzymes like PTP1B should cluster around the active forms of STEP.

RoPE torsional analysis operates on a shared set of torsion angles, including both main chain and all side chains. As the amino acid sequences of PTP1B and STEP are quite different (~30% sequence identity), this analysis would not fall within the capabilities of the current version of RoPE. This interpretation is consistent with the fact that all reported examples of RoPE usage involve many structures of only one protein at a time. However, we agree with the reviewer that future work would benefit from developing new analytical pipelines to compare ensembles of structures with sequence differences, which would enable exciting analyses in the future.

4. Line 476-479. The suggestion of a conserved allosteric hotspot across different PTPs is exciting!

We thank the reviewer for this kind comment! Indeed, the analyses we performed for this manuscript have proven powerful in increasing our understanding of the allosteric landscape of STEP, and in future work similar approaches will be applied in our study of other PTPs to ascertain similarities vs. differences between them.

5. Line 590. Given that the resolution cutoff was based on CC1/2, why is HiP CC1/2 (0.68) higher than the ones used for LoTP (0.374) and HiT (0.358)?

DIALS does indeed use CC1/2 as a basis for resolution cutoff, but not as the sole basis: CC1/2 is considered “within the context of autoprocessing in order to provide a consistent set of merging statistics for judging data during an ongoing experiment” (Gildea, et al., 2022). Of course, having high CC1/2 in the outer shell is not bad, as high CC1/2 indicates good merging. One could debate the merits of processing the data more aggressively to extend to slightly higher resolution. We also note that the other data processing statistics including $I/\sigma(I)$, R_{pim} , and completeness are similarly good for HiTP as for the other two datasets. We have amended the text in our methods to reflect this:

Resolution cutoffs were determined automatically by DIALS based on a combination of $CC_{1/2}$, $I/\sigma(I)$, R_{merge} , and completeness⁶².

6. Line 617. The 8SLT is cited twice; the last one should be 8SLU.

Thank you for bringing this to our attention. We have corrected this typographical error.

Reviewer #2 (Remarks to the Author):

This is a very interesting report contrasting structures of an enzyme obtained under contrasting thermodynamics conditions of temperature and pressure. The authors are correct to note that although the observed changes in structure are small, such small changes can have significant effects on catalysis. This reviewer found the RoPE analysis pretty convincing. Particularly interesting are the results obtained at loTHiP, conditions under which very few studies have been reported, despite solution studies revealing that pressure allows access to excited states of proteins and other biomolecules. Thus, these results are timely. In addition, as also noted by the authors, much insight remains to be gained from high temperature crystallography, and the present results support that notion.

We thank the reviewer for their kind remarks about the intriguing nature of our presented results, in particular with regard to the high-pressure data.

I have one point of disagreement with the interpretation of the results. The authors state: “... intriguingly, when subjected to pressure, the STEP protein molecule itself slightly expands, even as its environment is compressed. This differs from previous high-pressure crystal structures of 414 other proteins with a decreased protein molecular volume 27,29,31, but agrees with a high-pressure NMR solution structure with a slightly increased protein molecular volume, corresponding to negative compressibility 52. Our surprising, counterintuitive result suggests that the mechanisms by which pressure impacts the conformational landscapes of different proteins are complex and potentially context-sensitive.”

Actually pressure effects on conformational equilibria are due to differences in molar, not molecular volume. In many cases, pressure leads to more open, larger (in terms of molecular volume) forms of proteins, but the volume of the protein-solvent system (the molar volume) in these cases is nonetheless lower. Because in crystallographic studies, not all hydrating water molecules are

detected, it is not possible to estimate molar volume differences. Thus, in my view the observations are neither surprising nor counter-intuitive and need no negative compressibility to be explained. This said, more hydrated forms of proteins typically have lower compressibilities, so this is not contradictory. In any case, this interpretation should be revised.

We thank the reviewer for their comments regarding the effects of pressure on biomolecules, and particularly their insight into the effects on molar volume vs. molecular volume. Indeed, our observation that the protein molecular volume increases upon application of pressure is not surprising or counter-intuitive from a thermodynamic perspective when one considers the full protein crystal system including solvent, although it may be considered unintuitive (at least initially) to some readers. We have amended the relevant text to be more clear on these points, including omission of the words counter-intuitive, surprising, and negative compressibility:

In our structures, high temperature increases both unit cell volume and protein molecular volume (**Table 2**) as seen previously⁹. High pressure decreases unit cell volume as seen previously^{27,53,54}, yet still slightly increases protein molecular volume (**Table 2**). Thus, intriguingly, when subjected to pressure, the STEP protein molecule itself slightly expands, even as its environment is compressed. This differs from previous high-pressure crystal structures of other proteins with a decreased protein molecular volume^{27,29,31}, but agrees with a high-pressure NMR solution structure with a slightly increased protein molecular volume⁵⁵. **The slight increase in molecular volume of the protein itself that we observe upon pressurization may be initially unintuitive, but can likely be explained by counterbalancing decreases in the volume of the bulk solvent (which is invisible to crystallography) within the crystal lattice, and is thus consistent with the thermodynamic expectation that pressurization decreases the molar volume of the protein-solvent system. Nonetheless, our observations suggest that the mechanisms by which pressure impacts the detailed conformational landscapes of different proteins are complex and potentially context-sensitive.**

Along these lines, the convincing differences between pressure and temperature effects are consistent with differences in thermal expansion coefficients and molar volumes generally observed between states. However, these detailed structural studies show that these two thermodynamic perturbations lead to changes in distinct protein regions. This intricate insight is extremely interesting. Typically, large local compressibilities or changes in local conformation can be linked to the existence of nearby cavities which either compress, are filled with solvent or are eliminated by the conformational change induced by pressure. In contrast, internal void volume tends to increase with increasing temperature. It would have been nice to see a comparative analysis of the localization, size and shape of internal cavities in these three structures.

We agree with the reviewer that these observations are quite provocative! We also thank the reviewer for their comments about ways in which cavities could pertain to this analysis. To improve our manuscript in this regard, we have performed an analysis of the changes in the volumes of two specific cavities/pockets at key sites (the allosteric activator binding site and the active site), as well as an overall analysis of the changes in the distribution of cavity/pocket volumes throughout the entire

protein for each structure. We have thus added the following text (within a new Results section) as well as a new supplementary figure (Supp. Fig. 10) to our paper:

Effects on pockets and cavities

To assess how temperature vs. pressure affect various packing defects in the structure of STEP, we used the program CASTp⁵⁰ to measure the volumes of all pockets and/or cavities in each structure (**Supp. Fig. 10**). Interestingly, the largest pocket in each of the three structures was the allosteric activator site. Relative to the LoTP structure (128.7 Å³), the volume of this pocket increased at HiT (140.0 Å³) but decreased at HiP (74.8 Å³), consistent with general expectations of expansion with increasing temperature and compression with increasing pressure.

Considering all pockets/cavities in each structure, the mean volume relative to LoTP (7.8 Å³) increases for HiT (11.3 Å³) and slightly decreases for HiP (7.5 Å³). However, relative to LoTP, we observe no statistically significant difference in the distribution of these volumes for either HiT or HiP (Welch's two-sample t-test, p=0.951 and p=0.951, respectively). Thus, although some individual pockets react differently to different perturbations, at least by some measures the overall packing in the protein is not dramatically different.

Supplementary Figure 10: Distribution of pocket/cavity volumes for each structure.

Histograms of solvent-accessible pocket volume from CASTp⁵⁰.

- a) LoTP.
- b) HiT.
- c) HiP.

REVIEWERS' COMMENTS:

Reviewer #1 (Remarks to the Author):

The manuscript describes the use of high pressure (205 MPa) and physiological temperature (310 K) during protein single crystal X-ray diffraction data collection to expose new conformations of STEP (PTPN5), a clinically relevant protein tyrosine phosphatase. The present work is valuable for the reemerging field of room temperature crystallography and even more important for the less explored high-pressure experiments. The authors have addressed the concerns raised and rectified the molecular models accordingly.

Reviewer #2 (Remarks to the Author):

The authors have responded appropriately in my view to my comments. It also appears that they have addressed the comments of the other reviewer as well. My recommendation is to accept this revised version